# Regional response of grassland productivity to changing environment conditions influenced by limiting factors

**Qiuyue Li[1], Jihua Hou[1]\*, Pu Yan[2,3], Li Xu[2], Zhi Chen[2], Hao Yang[2], Nianpeng He[2,3,4]\***

**1** School of Ecology and Nature Conservation, Beijing Forestry University, Beijing, China, **2** Key Laboratory of Ecosystem Network Observation and Modeling, Institute of Geographic Sciences and Natural Resources Research, CAS, Beijing, China, **3** College of Resources and Environment, University of Chinese Academy of Sciences, Beijing, China, **4** Key Laboratory of Vegetation Ecology, Ministry of Education, Northeast Normal University, Changchun, China

\* houjihua@bjfu.edu.cn (JH); henp@igsnrr.ac.cn (NH)

**Data Availability Statement:** All relevant data are available from Dryad (DOI: 10.5061/dryad. 8sf7m0ckg).

## Abstract

Regional differences and regulatory mechanisms of vegetation productivity response to changing environmental conditions constitute a core issue in macroecological researches. To verify the main limiting factors of different macrosystems [temperature-limited Tibetan Plateau (TP), precipitation-limited Mongolian Plateau (MP), and nutrient-limited Loess Plateau (LP)], we conducted a comparative survey of the east-west grassland transects on the three plateaus and explored the factors limiting regional productivity and their underlying mechanisms. The results showed that aboveground net primary productivity (ANPP) of LP ($109.10 \pm 16.76$ g m$^{-2}$ yr$^{-1}$) was significantly higher than that of MP ($66.71 \pm 11.11$ g m$^{-2}$ yr$^{-1}$) and TP ($57.02 \pm 10.59$ g m$^{-2}$ yr$^{-1}$). The response rate of ANPP with environmental changes was different among different plateaus, being closely related to the main limiting factors. On MP, this was precipitation, on LP it was temperature and nutrients, and on TP, it was non-specific, reflecting restriction by the extremely low temperature. After autocorrelation screening of environmental factors, different regions exhibited different productivity response mechanisms. MP was mainly influenced by temperature and precipitation, LP was influenced by temperature and nutrient, and TP was influenced by nutrient, reflecting the modifying effect of the main limiting factors. The effect of each regional environment on ANPP was 72.56% on average and only 27.18% after simple regional integration. The regional model could optimize the simulation error of the integrated model, and the relative deviations in MP, LP, and TP were reduced by 31.76%, 17.22%, and 2.23%, respectively. These findings indicate that the grasslands on the three plateaus may have different or even the opposite mechanisms to control productivity.

## Introduction

Vegetation productivity, the productive capacity of plant communities under natural environmental conditions, is a research hotspot in terrestrial ecosystems [1]. The most direct

**Funding:** We received support for this research from National Natural Science Foundation of China, 31961143022, 31870437, National Key R&D Program of China, 2017YFA0604803. The funders had no role in study design, data collection and analysis, decision to publish, or preparation of the manuscript.

**Competing interests:** The authors have declared that no competing interests exist.

manifestation of vegetation productivity is food and fuel, which are closely related to human survival. It is estimated that approximately 40% of vegetation productivity in terrestrial ecosystem can be directly or indirectly utilized by humans [2]. Therefore, improving the simulation accuracy and forecasting ability of vegetation productivity models is of great significance for evaluating for ecosystem carrying capacity and sustainable development of the ecological environment [3].

Widespread regional variation is one of the major challenges for estimating large-scale vegetation productivity, and it is a common problem faced by ecologists. For estimating vegetation productivity at regional and global scales, model simulation is the most informative method, while field surveys and observations often verify the simulation accuracy of the model [4]. In previous studies, the estimations of vegetation productivity mainly focused on improving the universality of a model, making the relationship model show trends in parameter enrichment and structural complexity [5, 6]. As a combination of different geographic plates or biota, Earth's surface is influenced by factors such as topography, altitude, and distance from the ocean. Therefore, each region should have different environmental regulatory mechanisms and show completely different characteristics at different spatial scales or in different seasons [7, 8]. Therefore, in a unified empirical model obtained from certain biota or global data, the accuracy error of regional simulation needs to be further explored or quantified.

On different plateaus (or macrosystems), the role of major limiting factors in the response mechanism of vegetation productivity to changing environmental conditions may be an important theoretical basis for solving such problems. In natural ecosystems, based on Liebig's law of "minimal factors" [9], there is always one factor that reaches a state of insufficiency first, leading to the stability in entire system. Thus, limiting factors at the regional scale may be considered as relatively insufficient ecological factors after regional comparison. In addition to the regulation of various environmental factors, the level of vegetation productivity is also closely related to plant attributes [10, 11]. However, considering the interaction and co-evolution between plant attributes and main regional ecological factors [12], the concept of regional limiting factors is worth paying more attention to. At this level, these factors differ from the limiting factors at individual or population levels and are based on the overall control of the biota and large-scale environment. Therefore, we can compare regions with different main limiting factors to compare the response intensity at which vegetation productivity responds to environmental changes and the regional variation in response mechanisms to assess the status of regional limiting factors in the general promotion of large-scale productivity models.

Generally, research based on limiting factors has mostly been carried out under controlled experiments, but research using natural global change transects with evidently different limiting factors are rare. The global change terrestrial transect [13] is arranged along the direction of change for the main or secondary driving factors, such as temperature, precipitation, land use intensity, and nutrient status [14]. The transect has the characteristic of "replace time with space", in that regional spatial changes of the gradient can be regarded as a long-term ecological change. To some extent, these transects can be understood as long-term control experiments preset by earth; the biggest difference between them and control experiments is that the former reflect long-term adaptations in plants, whereas the latter focus on short-term responses. Therefore, experimental design on the concept of terrestrial transects is ideal for exploring the response mechanism of vegetation productivity under different regional limiting factors.

Temperature, precipitation and nutrients are important drivers of global change. Many control experiments have shown that extreme temperatures [15] and droughts [16] will significantly reduce aboveground net primary productivity (ANPP), and the synergy of N and P [17] will promote ANPP. The grasslands of the Tibetan Plateau (TP), Loess Plateau (LP) and

Mongolian Plateau (MP), as the specific macrosystems in the Northern Hemisphere, may be ideal locations for verifying the effects of regional limiting factors. On the TP, owing to high altitude and widespread glaciers, low average temperature may be the main factor limiting plant growth [18, 19]. On the LP, owing to its unique geological structure and topography formed by aeolian soil, soil erosion is serious, meaning that a lack of nutrients may be the main factor limiting vegetation growth [20, 21]. The MP is an arid and semi-arid region, and the grassland vegetation dynamics are related to the variability in precipitation, and thus, insufficient precipitation is the main limiting factor [22].

Based on the new idea of comparative transect, the present study focuses on the grasslands of the TP, LP, and MP to explore the regional response of productivity to environmental changes and to verify the modification effects of the main limiting factors. Our research was intended to determine the following: 1) distribution patterns of vegetation productivity in three typical grassland macrosystems; 2) environmental response characteristics and specific expression of vegetation productivity in different macrosystems; and 3) main mechanisms underlying grassland productivity in different macrosystems. Further verify the hypothesis that the regional limiting factor plays a leading role in the productivity response mechanism.

## Methods

### Study area

Typical grassland ecosystems on three plateaus in the Northern Hemisphere were selected (31–45˚N, 80–123˚W; S1 Table). The three plateau transects were intended to represent regions restricted by temperature, precipitation, and nutrients, and the measured data for mean annual temperature (MAT) on the TP, mean annual precipitation (MAP) on the MP, and soil N content on the LP support this inference (Fig 1). The map data illustrated in Fig 1 was derived from Land Cover Climate Modeling Grid product (MCD12C1) (https://lpdaac. usgs.gov/products/mcd12c1v006/).

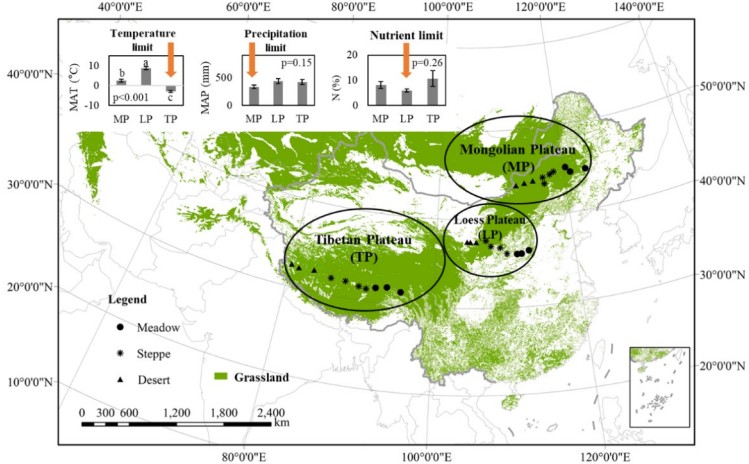

**Fig 1. Grassland distribution in the Northern Hemisphere and contrastive grassland transect between the Mongolian Plateau, loess Plateau, and Tibetan Plateau.** After comparing the three regions, the main limiting factors of each region were obtained. Grasslands on the Tibetan Plateau have a relatively low mean annual temperature (MAT), which is considered to lead to temperature limitation; grasslands on the Mongolian Plateau have a relatively low mean annual precipitation (MAP), which is considered to lead to precipitation limitations; grasslands on the Loess Plateau have relatively low total soil N leading to nutrient limitation.

The average altitude of TP is >4000m, and MAT is <0˚C, the highest monthly average temperature in <10˚C, and the MAP is 20–487mm [23]. There are three grassland types, namely, alpine meadow, alpine grassland, and alpine desert from southeast to northwest [24]. For LP, the altitude is 300–3000 m, the MAT is 3.7–14.0˚C, and the MAP is ~110–860 mm, and this plateau belongs to the dry with cold semi-arid climate (Bsk) and snow with dry winter climate (Dwa) [25–27]. The vegetation types were distributed from the southeast to northwest with warm forest, warm forest grassland, warm typical grassland, and warm desert grassland [28]. The MP, in a cold semi-arid climate (Bsk) [27, 29], is within a typical temperate continental climate (Dwb), with an MAT of −1.7–5.6˚C and an MAP of 90–433 mm [30]. From east to west, there are forests, forest grasslands, meadow grasslands, typical grasslands, desert grasslands and deserts [31].

## Transects setup

Grassland transects across the TP, LP, and MP were spread out along the precipitation gradient. There were 10 sites set up from east to west on each plateau. Sites 1–3 were in meadows, sites 4–7 were in steppes, and sites 8–10 were in deserts (Fig 1, S1 Table). All sites for grassland investigation were selected from natural grasslands with little human activity or grazing. To enhance the comparability among the three transects, the classification of grassland vegetation types was relatively simple (S1 Fig), which differed from the professional vegetation grassland classification system that emphasizes differences among vegetation groups in different regions [32, 33]. Two 50-m paralleled splines within the site were setup as repetitions, with four plots evenly arranged within each spline.

The transect on the TP spanned ~1600 km at an altitude of 4104–4938 m. The transect on the LP spanned >800 km at an altitude of 804–1714 m. The transect on the MP spanned >900 km at an altitude of 144–1272 m. (No specific permission was requirement for these locations to conduct field investigation for the aim of natural science in China, because these lands are public and these investigations did not involve endangered species.).

## Field survey

The field investigation was carried out in the peak plant growth period from July to August. In each plot (1 m × 1 m), we first collected litter and standing litter. Then we estimated the total coverage and average height and measured the plant height, sub-coverage, and abundance of species. Finally, we collected the aboveground parts of different plant species. A total of 260 species, 152 genera, and 48 families were collected. The samples were oven dried at 85˚C and to a constant weight to calculate the aboveground biomass (AGB). Soil samples were also collected using soil drills from each plot. After air-drying at 25˚C, we removed plant roots, gravel, and other debris, passed the samples through a 2-mm soil sieve, and ground them using a ball mill (MM400, Retsch, Haan, Germany).

## Data sources

**Aboveground net primary productivity.** AGB was obtained through plot survey in the late growing season of the grassland. For herb plants, AGB was considered as ANPP, and for shrub plants, we use the linear equation from Chen et al. [34]:

$$ln(ANPP) = b \times ln(AGB) + a \tag{1}$$

$$ANPP = a \times AGB + b \tag{2}$$

where both *a* and *b* are constants, and the constant values are different in different regions or shrub communities. This series of equations did not consider the shrub age, which will lead to the underestimation of ANPP.

**Climate data.** The climate data was extracted from online datasets based on the longitude and latitude. The MAT and MAP data were derived from the National Earth System Science Data Center, National Science & Technology Infrastructure of China (http://www.geodata.cn). The Aridity data came from the Global-Aridity and Global-PET Database [35] of Consortium for Spatial Information of the Consultative Group on International Agricultural Research (https://cgiarcsi.community/data/global-aridity-and-pet-database/).

**Nutrient data.** Soil nutrient data are obtained from the actual measurement of soil samples. The soil total N was measured by elemental analyzer (Vario MAX CN, Elementar, Germany), and the soil total P and soil total K were measured by a microwave digestion system (MARS Xpress, CEM, Matthews, USA) and an inductively coupled plasma emission spectrometer (ICP-OES, Optima 5300 DV, Perkin Elmer, Waltham, MA, USA).

## Data analysis

For the geographical distribution of ANPP, multiple comparisons (Duncan's test, $\alpha = 0.05$) were used to test the significance of ANPP differences in different regions and grassland types. Ordinary least squares was used to check the response rate (intensity) of ANPP to environmental factors in various regions. Furthermore, standardized major axis analysis was used to test the significance of the difference in response rates between different regions. We used stepwise regression to obtain the main master model of ANPP, and then compared the simulation deviation caused by simple integration. The initial model included MAT, MAP, Aridity, N, P, K parameters. On the basis of the species productivity matrix of sites, canonical correspondence analysis was used to interpret the ANPP based on environmental factors. Data analysis was performed by R-3.5.2 [36–38], and charts were drawn in PowerPoint 2010 and R-3.5.2. The significance test level was $P < 0.05$.

## Results

### Spatial variation in grassland ANPP

The grassland ANPP of three plateaus ranged from $14.44 \pm 2.96$ g m$^{-2}$ yr$^{-1}$ to $175.16 \pm 99.87$ g m$^{-2}$ yr$^{-1}$. The ANPP of each transect showed a significant upward trend with longitude from west to east (Fig 2a). The average of ANPP on LP was $109.10 \pm 16.76$ g m$^{-2}$ yr$^{-1}$, which was significantly higher than that on the MP ($66.71 \pm 11.11$ g m$^{-2}$ yr$^{-1}$) and TP ($57.02 \pm 10.59$ g m$^{-2}$ yr$^{-1}$) (Fig 2b). Among different grassland types, the ANPP on all plateaus showed the following similar trend: meadow > grassland > desert (Fig 2c).

### Regional specificity of grassland productivity response to climate and nutrient changes

The response of ANPP to changes in climate (Fig 3a) and nutrients (Fig 3b) were mostly positively correlated, reflecting that high temperature and humid promote productivity. Furthermore, there were significant differences in the intensity of vegetation productivity response to environmental factors in different regions (Fig 4), and the response was closely related to the main limiting factors in each region.

The response intensity of ANPP with MAP on the MP (0.431) was significantly higher than that on the LP (0.235) and TP (0.197). Therefore, MP had a stronger precipitation response specificity, corresponding to precipitation limitation in this region. Owing to the limitation of

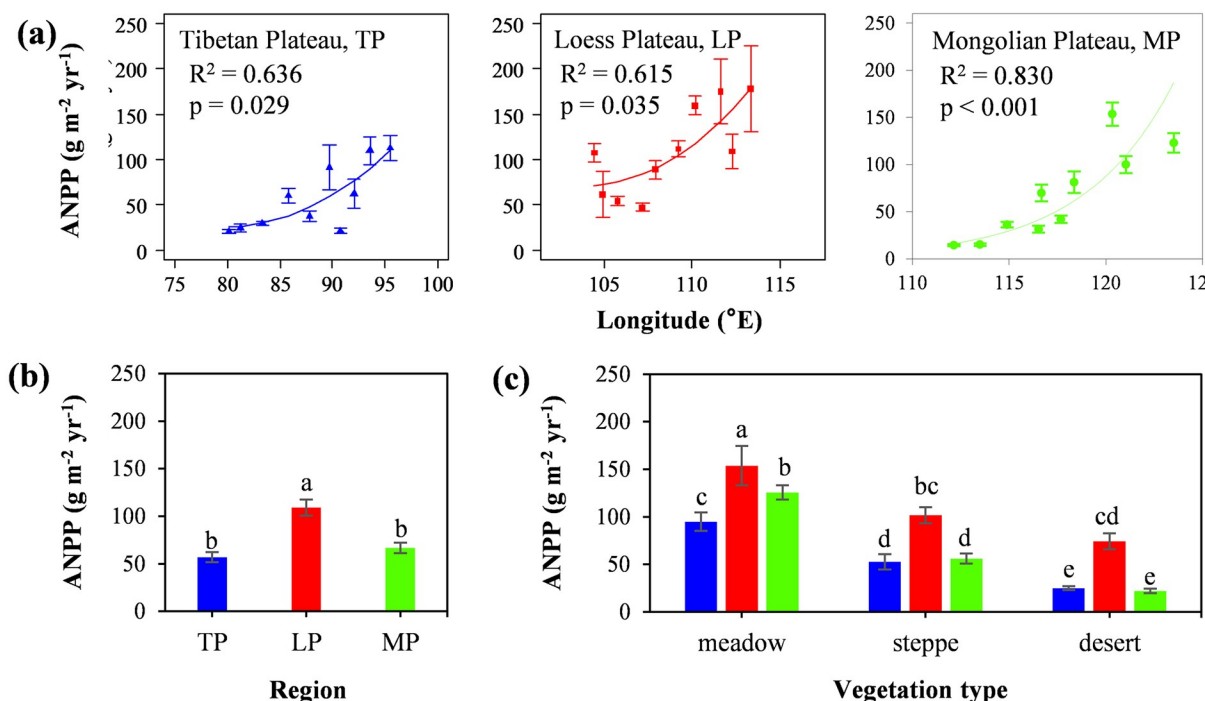

**Fig 2. The distribution pattern of grassland aboveground net primary productivity (ANPP) between Mongolian Plateau, loess Plateau, and Tibetan Plateau.** The error line is one times the standard error. Different letters (a, b, c, d, e) indicate significant difference ($p < 0.05$).

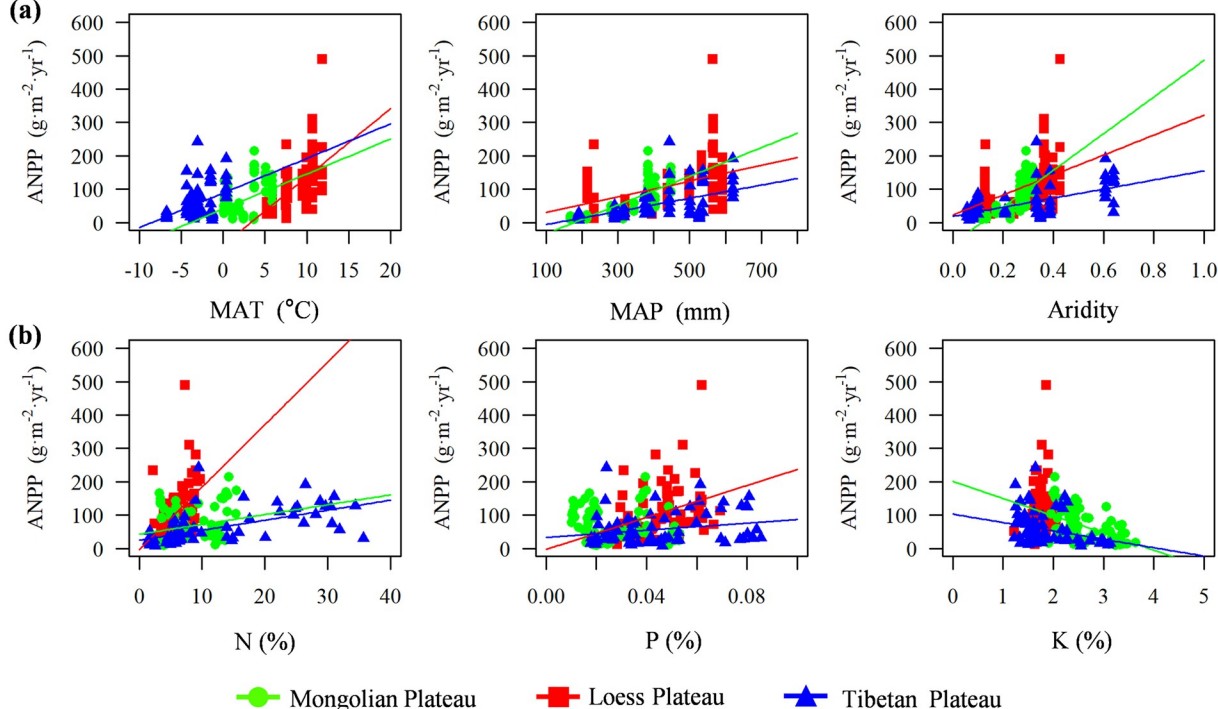

**Fig 3. The relationship of aboveground net primary productivity (ANPP) to climate factors and soil nutrient among different regions.** Solid line shows that ANPP was significantly correlated with environmental factors ($p < 0.05$). MAT, mean annual temperature; MAP, mean annual precipitation.

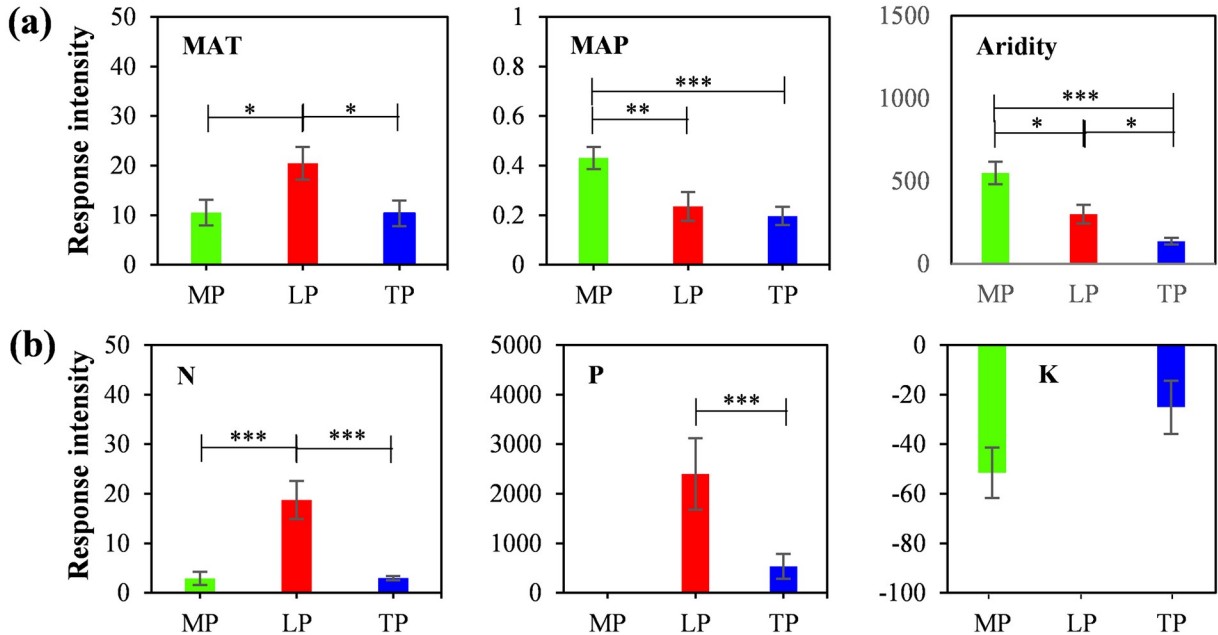

**Fig 4. Changes in the response intensity of aboveground net primary productivity (ANPP) to climate factors and soil nutrients among the different regions.** The response intensity is the slope of the linear fitting equation between ANPP and environmental factors; the error line represents one times of standard error; * represents the significance of the two regions, *p < 0.05; **: p<0.01; ***: p < 0.001. MAT, mean annual temperature; MAP, mean annual precipitation.

excessively low temperature, the response of ANPP on the TP to the environment was overall low, which in turn leads to its non-specificity temperature response. The LP showed a strong response specificity in MAT, N and P, and especially N, indicating that the changes in ANPP were closely related to nutrient limitation.

## Comprehensive regulation of environmental factors on grassland productivity modified by limiting factors

After filtering the autocorrelation factors, each regional environmental factor presented different models to controlling ANPP (Table 1). When the total data from the three regions were considered, ANPP showed a master model of MAT, MAP, and N, reflecting the overall regulation of temperature, precipitation, and nutrients. After the regions were screened, the main master model of each region was different and was closely related to the main limiting factors

**Table 1. The master model of aboveground net primary productivity (ANPP) controlled by environmental factors in different regions.**

| Region | Model | Adj.R$^2$ | p |
|---|---|---|---|
| Mongolian Plateau | ANPP = -88.54910* + 8.22961 MAT + 0.40518** MAP | 0.7105 | 0.0054 |
| Loess Plateau | ANPP = -92.659* + 14.114* MAT + 6.060 N + 927.095 P | 0.8915 | <0.001 |
| Tibetan Plateau | ANPP = 23.1177* + 3.1529** N | 0.7184 | 0.0012 |
| Total | ANPP = -17.71743 +5.51664***MAT + 0.17237** MAP + 1.47000 N | 0.7046 | <0.001 |

Initial factors include: MAT, mean annual temperature; MAP, mean annual precipitation; Aridity; N; P; and K

* represents the significance of the regression coefficient, *p < 0.05;

**: p<0.01;

***: p < 0.001

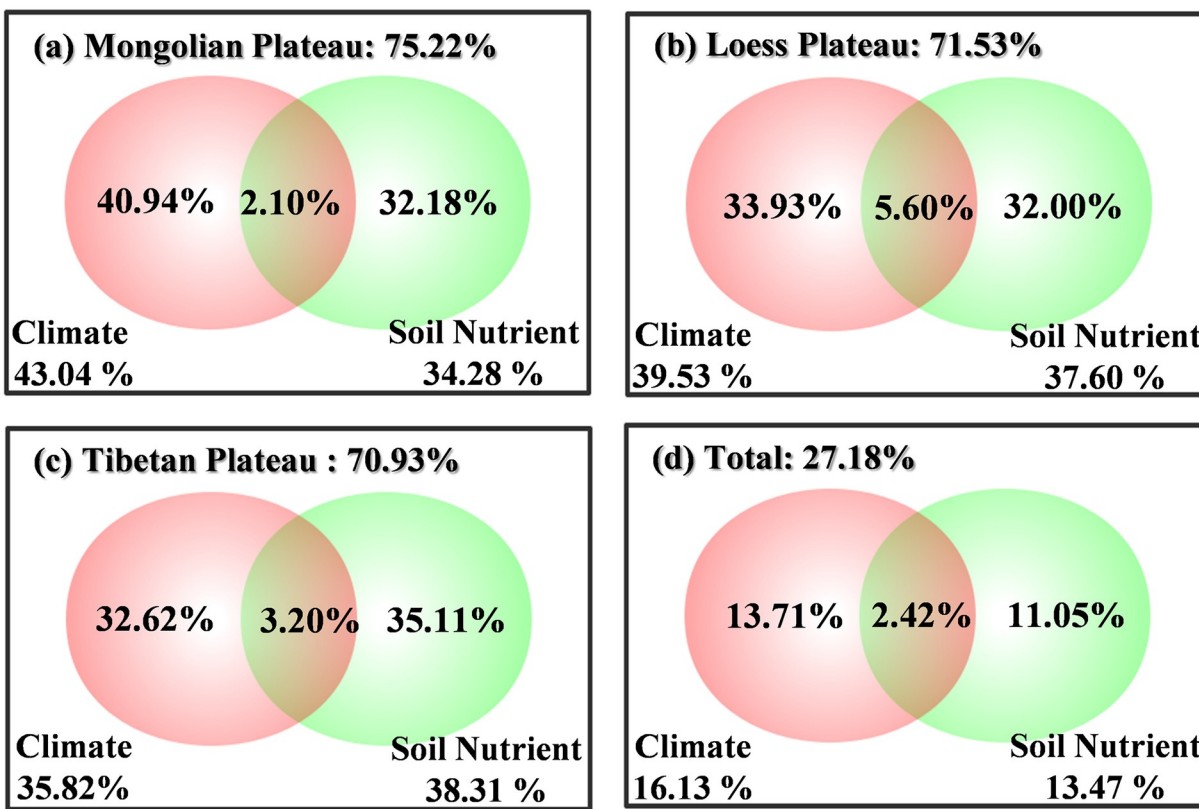

**Fig 5. Environmental contribution of grassland aboveground net primary productivity (ANPP) among different regions by canonical correspondence analysis.** Climate factors were mean annual temperature (MAT), mean annual precipitation (MAP), and Aridity, and soil nutrients were N, P, and K. The response variable was the biomass matrix of the species, and the explanatory variable was the environmental factor matrix.

on each plateau. The ANPP of the MP was comprehensively regulated by MAT and MAP, but on the LP, ANPP was regulated by MAT, N, and P, and on the TP, ANPP was regulated by N.

### Difference between regional specificity and simple regional integration

In each region, the contribution of climate factors and soil nutrients to grassland ANPP was 72.56% on average. This wad 75.22% on the MP (Fig 5a), 71.53% on the LP (Fig 5b), and 70.93% on the TP (Fig 5c). After simply integrating the data of the three regions, the contribution of climate and soil nutrient factors was only 27.18% (Fig 5d), indicating that the three plateaus may have different or even divergent productivity response to changing environment conditions.

Comparing the fitting errors of the integrated model and the regional model (Fig 6a), the absolute and relative deviations of the regional model were considerably lower than those of the integrated model (Fig 6b), especially on the LP. Among them, the relative deviation of the MP, LP and TP decreased by 31.76%, 17.22%, and 2.23%, respectively.

## Discussion

### Regional variation in grassland ANPP

Within each plateau, ANPP showed a gradual upward trend as the longitude increased (Fig 2). Comparatively, ANPP first increased and then decreased as TP–LP–MP with the increase in

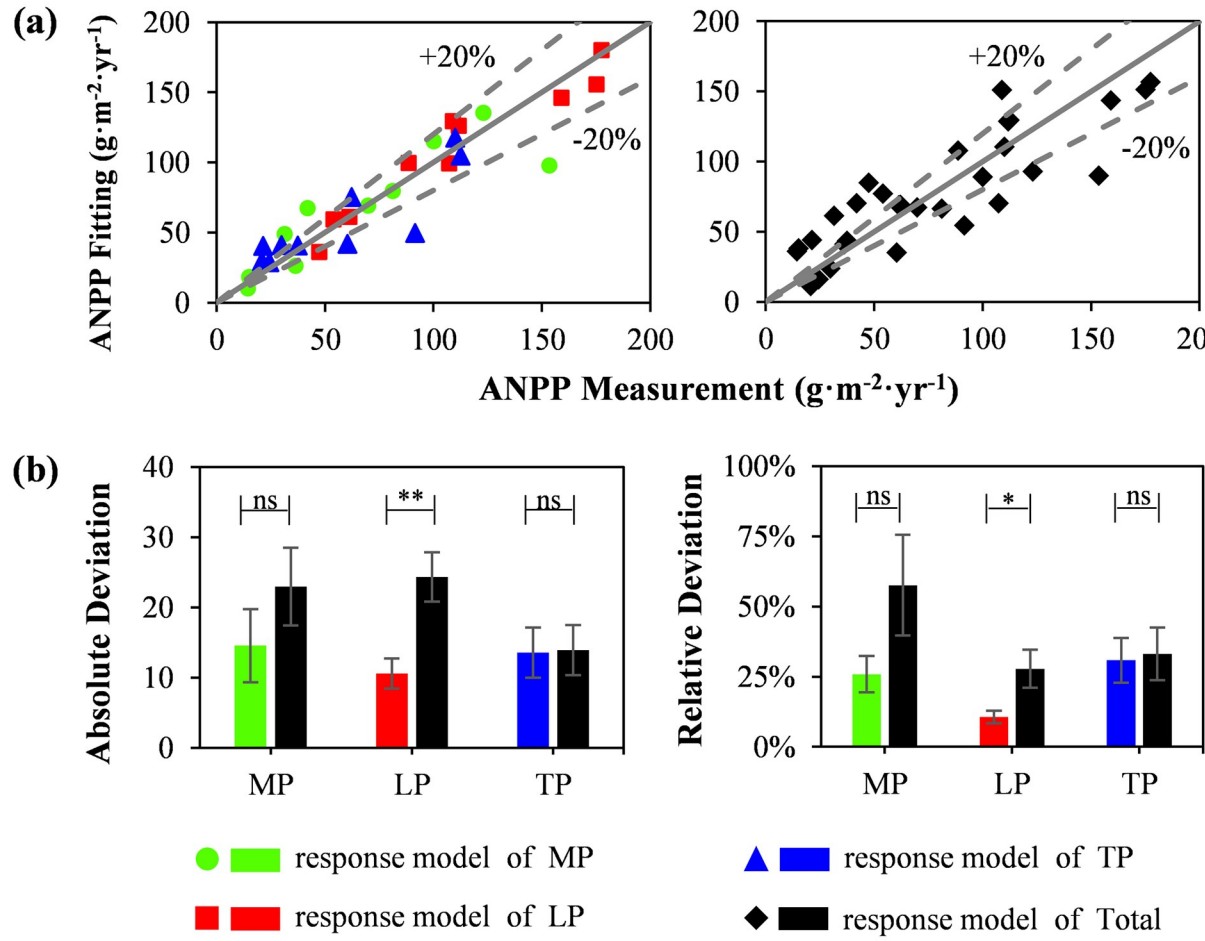

**Fig 6. Fitting comparison between aboveground net primary productivity (ANPP) of the regional model and integrated model.** Dotted line ± 20%, indicating the range of ANPP measured value fluctuation by 20%; * represents the significance of the two regions, *p < 0.05; **: p<0.01; ***: p < 0.001; ns, no significant difference. MP, Mongolian Plateau; LP, Loess Plateau; TP, Tibetan Plateau.

latitude. These results agreed with those reported by Jiao et al. [39] in Europe. Moreover, the change trend of ANPP along the transect was consistent with the regional vegetation zonality, which demonstrated that the design of the transect can successfully obtain the characteristics of regional vegetation.

Compared with that on the TP and MP, the grasslands of ANPP on the LP was highest. These results verified that, compared with nutrient limitation, temperature and precipitation limitations have a greater effect on vegetation productivity on the TP and MP. The growth inhibition caused by nutrient deficiency is more common in trees or shrubs [20] compared with herb, so grassland shows higher ANPP. The grasslands on the MP were mostly affected by the extreme arid climate [30], but precipitation extremes have declined in recent years [40]. The water stress due to extreme drought can easily result in water imbalance in grasslands [40]. Most surviving plants present resource-conservative functional traits [41], such as higher growth of underground roots [42] or an earlier-ending growing season [43], resulting in lower ANPP. The extreme low temperature on the TP is a long-term stress factor, and the low productivity is understandable. Low temperature can inhibit the activity of plant cell enzymes, resulting in slower plant growth and limited organic matter accumulation during the short period in which the soil thaws [44]. In addition, the alpine vegetation on the TP is more dwarf

[45], and grows in a unique high-density "straw felt" pattern, allowing plants to gather together for warmth.

## Specific performance and response mechanisms of ANPP to environmental changes modified by the main limiting factors

It is important to explore the response mechanism of ANPP to global change, and regional limiting factors may be the key to understanding the productivity response mechanism [46, 47]. Regional characteristics can be showed by comparing large-scale transects, and the regional vegetation productivity regulatory mechanisms maybe not easily change. For example, along these transects, ANPP had significant linear relationships with environmental factors, irrespective of the plateaus (Fig 3), although the response intensity of grassland ANPP in different regions was also significant different (Fig 4). Among them, the response intensity of ANPP with changes in K was not significant, although there were differences among region [48].

On the MP, the response intensity of ANPP with changes in MAP was much higher than that on the LP and TP. Previous studies have demonstrated that precipitation is the main factor affecting ANPP in arid and semi-arid regions [49], and semi-arid grasslands are highly sensitive to fluctuation in precipitation [50]. After filtering the autocorrelated factors, ANPP was comprehensively regulated by the MAT and MAP (Table 1), being specified by aridity (Fig 4). Temperature is important for enzyme activity to promote plant photosynthesis [51] and more efficiently utilize precipitation and soil nutrients.

On the LP, grassland ANPP had a specific response to temperature and nutrients (Fig 4), and was comprehensively regulated by MAT, N, and P (Table 1). Owing to severe soil erosion and nutrient loss on this plateau [52], plants must rapidly respond to changing nutrients. The vegetation of LP was considered to be close to the threshold of regional water resources carrying capacity [53]. Compared with grassland, forest was the main body that affects the balance of water use [54]. Therefore, the precipitation limitation of grassland vegetation may not be strong. Temperature may directly affect plant metabolism and transpiration [55], further influencing the rate of photosynthesis and the absorption of water and nutrients by roots. Furthermore, the sensitivity to temperature can alleviate the growth limitation due to nutrient deficiency [28].

On the TP, the response intensity of ANPP to environmental changes was overall low (Fig 4), reflecting the overall suppression of plant growth under extremely low temperatures [56]. As the important water source of China and even Asia, the TP is not water limited [57, 58], and soil nutrients are mostly stored in an organic state [59]. However, too low temperature may depress soil N mineralization, resulting in an apparent limitation of available N [60]. When autocorrelated factors were filtered, the strong regulation of soil N content was shown (Table 1). Under long-term low temperature stress, alpine plants have evolved a variety of adaptive strategies, such as dense villi and stolon or mat-like growth [61]. This shows that, under long-term adaptation, cold-tolerant herbs grow well on TP and have formed stable plant physiological characteristics. Although the grasslands of the TP are more sensitive to warming, at the regional scale, the vegetation rejuvenation period has not significantly advanced [62] and the optimal length of the growing season has shortened [23]. Moreover, the warming and drying trend in the western region [63] have no significant effect on grassland productivity.

## Regional limiting factors should be emphasized during regional integration

In a simple integration of different regional data, the effect of environmental factors on ANPP greatly decreased (Fig 5), and the fitting bias of the simple model increased (Fig 6). This

evidence reveals that differences in environmental regulatory mechanisms are common among different macrosystems. Owing to the differences in the environmental conditions of different macrosystems, a simple model is not sufficient for reflecting the whole system, and regional limited factors should be emphasized. Therefore, to efficiently estimate productivity, we not only need to judge regional characteristics [6, 64–66] to establish a model but also to consider more information regarding the key limiting factors on, e.g., regression trees and neural networks.

We are far from identifying the main limiting factors at the regional level because few studies have been reported on this subject. A more complete theoretical foundation is needed to further discuss the temporal and spatial scale of limiting factors. Furthermore, how to determine the main limiting factors at regional scale is dependent upon the environmental parameters collected in a specific study. In the present study, data on regional vegetation, being simply reflected in the transect survey, may have inherent errors due to the selection of sites and the influence of investigation time. In practice, the setup of the plateau transect basically follows the existing transect proposed and established by previous researchers [67, 68]. The present study is the first, to our knowledge, to attempt to systematically compare these transects. In the future, more systematic transect surveys can be carried out using a consistent protocol, even covering different fields (e.g., plants, animals, microorganisms, and soil) and different data collection scales (e.g., ground, remote sensing, lidar, models, and flux).

## Conclusion

There are significant regional differences in the response of grassland productivity to changing environment conditions, and the main limitation factors in different regions can modify the regulatory mechanisms. The response of ANPP to changing environments on the LP, MP and TP was mostly related to the specific limiting factors in each region but has different mechanisms driving the response rate and direction. When using the model to simulate grassland ANPP at a large scale, the regional limiting factor, as a breakthrough point, should be emphasized to improve its simulation accuracy. In future, the comparative transects are not only ideal for exploring the response mechanism of productivity but also represent a research platform for multidisciplinary integration (e.g., plants, animals, and microorganisms). This is also expected to be important for the verification of regional limiting factors.

## Supporting information

**S1 Fig. Difference of drought degree of grassland types in Mongolia Plateau, Loess Plateau and Tibetan Plateau.** Error line represents 1 * standard error; different letters (a, b) indicate significant difference (P < 0.05).
(TIF)

**S1 Table. The basic information of the field-investigated.**
(DOCX)

## Acknowledgments

We would like to thank these stations from China Ecosystem Research Network (CERN). In field investigation, Lhasa Plateau Ecological Research Station, Ansai Research Station of Soil and Water Conservation, and Inner Mongolia Grassland Ecosystem Research Station provide important logistics support and botanical expertise.

## Author Contributions

**Conceptualization:** Jihua Hou, Nianpeng He.

**Data curation:** Qiuyue Li, Jihua Hou, Pu Yan, Nianpeng He.

**Formal analysis:** Hao Yang, Nianpeng He.

**Funding acquisition:** Hao Yang, Nianpeng He.

**Investigation:** Pu Yan, Li Xu, Zhi Chen, Hao Yang.

**Methodology:** Jihua Hou.

**Writing – original draft:** Qiuyue Li, Li Xu, Zhi Chen, Nianpeng He.

**Writing – review & editing:** Jihua Hou, Nianpeng He.

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
