## [Decision Letter · Decision Letter 0]

16 Jul 2020

PONE-D-20-18058

Regional response of grassland productivity to changing environment influenced by limiting ecological factors

PLOS ONE

Dear Dr. He,

Thank you for submitting your manuscript to PLOS ONE. After careful consideration, we feel that it has merit but does not fully meet PLOS ONE’s publication criteria as it currently stands. Therefore, we invite you to submit a revised version of the manuscript that addresses the points raised during the review process.

We look forward to receiving your revised manuscript.

Kind regards,

Dafeng Hui, Ph.D.

Academic Editor

PLOS ONE

Journal Requirements:

2. In your Methods section, please provide additional location information of the study sites, including geographic coordinates for the data set if available.

3. In your Methods section, please provide additional information regarding the permits you obtained for the work. Please ensure you have included the full name of the authority that approved the study sites access and, if no permits were required, a brief statement explaining why.

5.Thank you for stating the following in the Acknowledgments Section of your manuscript:

[This work was supported by National Natural Science Foundation of China, No. 31870437, No. 31988102, the Chinese Academy of Sciences Strategic Priority Research Program (XDA19020302), National Key R&D Program of China (2017YFA0604803)；]

 [The funders had no role in study design, data collection and analysis, decision to publish, or preparation of the manuscript.]

7.We note that [Figure(s) 1] in your submission contain [map/satellite] images which may be copyrighted. All PLOS content is published under the Creative Commons Attribution License (CC BY 4.0), which means that the manuscript, images, and Supporting Information files will be freely available online, and any third party is permitted to access, download, copy, distribute, and use these materials in any way, even commercially, with proper attribution. For these reasons, we cannot publish previously copyrighted maps or satellite images created using proprietary data, such as Google software (Google Maps, Street View, and Earth). For more information, see our copyright guidelines: http://journals.plos.org/plosone/s/licenses-and-copyright.

1.    You may seek permission from the original copyright holder of Figure(s) [1] to publish the content specifically under the CC BY 4.0 license. 

Additional Editor Comments (if provided):

I now have two reports from expert reviewers. While reviewers consider the study important, they raised some serious concerns related to the technical part of the study. Reviewer #1 was more critical and had two major concerns, and recommended Reject. Both reviewers think English needs to be significantly improved. I concur with the reviewers and agree that the topic of the study is interesting, and would like to give the authors an opportunity to address the reviewers's concerns if they can.

Reviewers' comments:

Reviewer's Responses to Questions

**Comments to the Author**

1. Is the manuscript technically sound, and do the data support the conclusions?

Reviewer #1: No

Reviewer #2: Partly

2. Has the statistical analysis been performed appropriately and rigorously? 

Reviewer #1: No

Reviewer #2: Yes

3. Have the authors made all data underlying the findings in their manuscript fully available?

Reviewer #1: Yes

Reviewer #2: Yes

4. Is the manuscript presented in an intelligible fashion and written in standard English?

Reviewer #1: No

Reviewer #2: Yes

5. Review Comments to the Author

Reviewer #1: The paper compared the grassland productivity on the three plateaus (Tibetan Plateau, Mongolian Plateau, and Loess Plateau) and such differences related with environmental factors in those regions. The authors postulate that the main limiting factor in the three regions is low temperature for Tibetan Plateau, low precipitation (water) for Mongolian Plateau, and low soil nutrient for Loess Plateau. Although each place has its specific and primary environmental limiting factor, I do not think such classification is accurate. There are two main concerns: 1) the three plateaus are large, covering different climatic conditions, thus such single factor analysis cannot fully explain their specific site differences; 2) the environmental factors are interacted, and single factor cannot clearly indicate the mechanisms. The authors did lots of field survey and data analysis, while such study did not meet the judge standards in rationality and research integrity. Thus, I do not suggest it to be accepted for publication in Plos One.

Some specific comments are as following:

1. English writing needs improvement, and some sentences are vague.

2. Abstract: lack descriptions about experimental design and methods; the last sentence is redundant and can be deleted.

3. Introduction: some information about the three plateaus is not exact, such as limiting factors and grassland distribution (line 92-93).

4. Methods: more explanations about such descriptions in environmental limitations (line 116). It was normally considered that there are two main factors, i.e. water and nutrient, especially water on the Loess Plateau. Line: 125: This in not correct, some parts on loess plateau belong to arid region. Line:135-137: such classifications clearly show the rainfall or water gradient. Line156: such survey cannot represent the three plateaus as typically. Data analysis descriptor is not enough, and more detailed explanations such as those in Table 1 needed.

5. Results: line 207, such great differences among three plateaus implies that such comparisons cannot solely focus on environment factors. Line 220: how define the response density here? Line 233-235: such conclusion is incorrect. Line 239: suggest delete the sentence. Line 253: the title is unclear.

6. Discussion: generally, the discussion has poor relation with results obtained. The paper put environmental factors as fixed variable, while neglect their variations. Some conclusions lack of direct data results, while may be speculated (such as line 323).

7. Figure 1: there lack of statistical analysis for the three environmental factor limitations in columns. Figure 2: The ANPP increase exponentially with longitude on TP and LP? linealy on MP? correct or not? Figure 5: more information needed about the analysis?

8. Table 1: the results in the table conflict with the hypothesis, or expectations. The data was significant with whom as marked with *?

Reviewer #2: This study investigated the main regional limiting factors in the changes of grassland aboveground net primary productivity in Tibetan Plateau (TP), Mongolian Plateau (MP) and Loess Plateau (LP) by using a comparative transect survey dataset. This research topic is important as regional differences and regulation mechanism of vegetation productivity response to changing environmental is one of the core issues in macroecology research. To verify the main regional limiting factors, the authors used multiple analysis methods. However, analysis method usually has its own defects. For the analysis method of gradual regression, the significance depends not only on their intrinsic ecological implications but also on the variations of their selected samples. Why nutrient-N is the highest in TP, while the N relationship with TP productivity showed the best among the three regions. I think, at least partly, it is due to that the selected samples in TP have greater variations of N than those in the other two regions. Similarly, the variations of N in MP are much lower, thus it is difficult to develop a significant relationship between N and productivity. I suggest that the author fully acknowledged the limitations of this experimental design and clarified the potential methodological defects in the discussion. On the contrary, I like the results based on the canonical response analysis (CCA). When the data of the three regions are analyzed together, they found that the explanation degree of climate and soil nutrient factors reduced largely, indicating that the three plateaus may have different or even divergent productivity response mechanisms. I suggest that the author focus on this result for discussion. In addition, English writing could be improved. While the general writing was good, there were many Chinglish and long sentence that greatly affect the readability of this paper. For example, on Line 27, " MP is precipitation specific, LP is temperature and nutrient specific …"; Line 31" MP is mainly temperature and precipitation regulation, LP is temperature and nutrients, and TP is nutrients…". Lines 32-33, “The interpretation degree of environment to ANPP is 72.56% in the region, while only 27.18% after simple integration” Lines 341-345, “In the TP limited by low temperature, the response intensity…, which…,which…environmental factors in the TP.”

Based on the above reasons, I'd like to recommend a major revision before it can be accepted for publication.

6. PLOS authors have the option to publish the peer review history of their article (what does this mean?). If published, this will include your full peer review and any attached files.

Reviewer #1: No

Reviewer #2: No

---

## [Author Response · Author response to Decision Letter 0]

16 Sep 2020

Response to editor:

(Original comment and query in Bold, Response in upright Roman)

Response：Thanks for your comments. We have studied the format samples of PLOS ONE's literatures and adjusted the format of the title page and main body of this paper.

2. In your Methods section, please provide additional location information of the study sites, including geographic coordinates for the data set if available.

Response：Thanks for your suggestion. After this revision, the geographic coordinates of the study area will be added to the Method section: [The typical grassland ecosystems widely distributed in the northern hemisphere were selected as the research object (31–45°N, 80–123°W; Tab.S1).]. The specific latitude and longitude and basic climate information of the 30 sites involved can be viewed in Appendix 1 (Tab.S1).

3. In your Methods section, please provide additional information regarding the permits you obtained for the work. Please ensure you have included the full name of the authority that approved the study sites access and, if no permits were required, a brief statement explaining why.

Response：Thanks for your proposal. This research was carried out with the support of the Institute of Geographical Sciences and Natural Resources Research of the Chinese Academy of Sciences, relying on the field stations of the China Ecosystem Research Network. The full names of such institutions will be written in the acknowledgment, and will not be covered in the Method section.

Response：Thanks for your reminder. We do not need to change the Data Availability statement at this time

5.Thank you for stating the following in the Acknowledgments Section of your manuscript:

[This work was supported by National Natural Science Foundation of China, No. 31870437, No. 31988102, the Chinese Academy of Sciences Strategic Priority Research Program (XDA19020302), National Key R&D Program of China (2017YFA0604803)]

We note that you have provided funding information that is not currently declared in your Funding Statement. However, funding information should not appear in the Acknowledgments section or other areas of your manuscript. We will only publish funding information present in the Funding Statement section of the online submission form. Please remove any funding-related text from the manuscript and let us know how you would like to update your Funding Statement. Currently, your Funding Statement reads as follows: [The funders had no role in study design, data collection and analysis, decision to publish, or preparation of the manuscript.]

Response：Thanks for your proposal. After this revision, we have deleted the funding-related text from the manuscript and rewritten the Acknowledgment section to meet the requirements of your journal. 

The Acknowledgments section is revised as follows: [We would like to thank the Institute of Geographical Sciences and Natural Resources Research of the Chinese Academy of Sciences for substantial support throughout this research. Within the China Ecosystem Research Network, the Lhasa Plateau Ecological Research Station, Ansai Research Station of Soil and Water Conservation, and Inner Mongolia Grassland Ecosystem Research Station provide important logistics support and botanical expertise.] 

The Funding Statement is revised as follows: [we received support for this research from National Natural Science Foundation of China, No. 31870437, No. 31988102, the Chinese Academy of Sciences Strategic Priority Research Program (XDA19020302), National Key R&D Program of China (2017YFA0604803). The funders had no role in study design, data collection and analysis, decision to publish, or preparation of the manuscript.] 

Response：Thanks for your reminder. I am sure that the corresponding author in the Editor Manager has an ORCID iD and has been verified by the Editor Manager.

7.We note that [Figure(s) 1] in your submission contain [map/satellite] images which may be copyrighted. All PLOS content is published under the Creative Commons Attribution License (CC BY 4.0), which means that the manuscript, images, and Supporting Information files will be freely available online, and any third party is permitted to access, download, copy, distribute, and use these materials in any way, even commercially, with proper attribution. For these reasons, we cannot publish previously copyrighted maps or satellite images created using proprietary data, such as Google software (Google Maps, Street View, and Earth). For more information, see our copyright guidelines: http://journals.plos.org/plosone/s/licenses-and-copyright.

Response：Thanks for your reminder. The map data used in [Figure(s) 1] is freely available online to support any third-party scientific and educational purposes; and it complies with the relevant regulations of the CC BY 4.0 license. After this revision, the following description will be added to the method section: [The map data of Figure 1 comes from Land Cover Climate Modeling Grid (CMG) (MCD12C1) (https://lpdaac.usgs.gov/products/mcd12c1v006/)]

 

Response to reviews:

(Original comment and query in Bold, Response in upright Roman)

To reviewer 1:

The paper compared the grassland productivity on the three plateaus (Tibetan Plateau, Mongolian Plateau, and Loess Plateau) and such differences related with environmental factors in those regions. The authors postulate that the main limiting factor in the three regions is low temperature for Tibetan Plateau, low precipitation (water) for Mongolian Plateau, and low soil nutrient for Loess Plateau. Although each place has its specific and primary environmental limiting factor, I do not think such classification is accurate. There are two main concerns: 1) the three plateaus are large, covering different climatic conditions, thus such single factor analysis cannot fully explain their specific site differences; 2) the environmental factors are interacted, and single factor cannot clearly indicate the mechanisms. The authors did lots of field survey and data analysis, while such study did not meet the judge standards in rationality and research integrity. Thus, I do not suggest it to be accepted for publication in Plos One.

Response：Thanks very much for your critical comments above. We fully understand the serious concerns of reviewer about the technical part of the article, and have made corresponding rigorous revisions to the entire article. It is true that the types of microclimates in the plateau region are complex and changeable; and the interactions between environmental factors are difficult to decipher with simple models. We fully acknowledge the limitations of this experimental design and will clarify potential method flaws in the discussion. But the purpose of this research is to verify the possibility of deducing the main ecological limiting factors to a regional scale; and how well the limiting factors control the productivity response mechanism of the three plateaus. However, due to the time and manpower constraints of our research group, it is difficult to conduct grid-based, comprehensive field surveys on the three plateau regions. Although the setting of the terrestrial transect has its own shortcomings, it is also a better way to achieve the research purpose.

1. English writing needs improvement, and some sentences are vague.

Response：Thanks for the reviewer’s reminder. In this revision, we have revised the entire text one by one.

2. Abstract: lack descriptions about experimental design and methods; the last sentence is redundant and can be deleted.

Response：Thanks very much for the comments of the reviewers. This time, the Abstract has been completely revised; the description of the experimental design and methods has been renewed (Line:23-28), and the last sentence has been deleted.

3. Introduction: some information about the three plateaus is not exact, such as limiting factors and grassland distribution (line 92-93).

Response：Thanks for the reviewer's criticism. After this modification, related texts with inaccurate information have been corrected. Among them, clearly pointing out that the determination of regional limiting factors is only a reasonable inference based on Liebig's law in this article, not an established fact (Line:75-81). And whether the three plateaus are the main limiting factors for temperature, precipitation, and nutrients are also worthy of verification and discussion in this study.

4. Methods: more explanations about such descriptions in environmental limitations (line 116). It was normally considered that there are two main factors, i.e. water and nutrient, especially water on the Loess Plateau. Line: 125: This in not correct, some parts on loess plateau belong to arid region. Line:135-137: such classifications clearly show the rainfall or water gradient. Line156: such survey cannot represent the three plateaus as typically. Data analysis descriptor is not enough, and more detailed explanations such as those in Table 1 needed.

Response：Thanks for the reviewer's comment. According to your question, we have revised and rewritten the text of the method section: 1) We fully agree with the reviewer's statement that the vegetation on the Loess Plateau is restricted by moisture and nutrients. There are many studies on the water threshold of the Loess Plateau, but forests and shrubs are mostly selected as the research objects; this research mainly focuses on grassland vegetation, and the effect of water restriction may be slightly weakened. Relevant instructions have been added to the Discussion (line: 307-309, 343-345). 2) For the determination of the climate zone of the Loess Plateau, we have changed the "semi-arid and semi-humid area" to the "semi-arid continental monsoon climate zone" to ensure the rigor of the text (line: 145). 3) The layout of the plateau transects in this study is exactly along the east-west water-heat gradient especially the precipitation gradient; this description will be added after this revision (line:153-154). The same is true for the regional characteristics of the transects (Fig 2a). 4) We acknowledge that the experimental design of this study will have inherent errors in sites selection, and we will also add corresponding explanations in the Discussion section (line:383-386). Although the transect survey can only simply characterize the general level of each region, the selection of sample points has been done through many literature references and the team’s early field investigations, which has tried to satisfy the objectivity and scientificity of the experiment. In the future, if we can get more time and financial support, we will be able to set up sites in a grid to reflect regional characteristics more truly. 5) The text of data analysis is a description of the process and method of drawing all charts; corresponding expanded instructions have been carried out.

5. Results: line 207, such great differences among three plateaus implies that such comparisons cannot solely focus on environment factors. Line 220: how define the response density here? Line 233-235: such conclusion is incorrect. Line 239: suggest delete the sentence. Line 253: the title is unclear.

Response：Thanks for the reviewer's criticism. 1) The reasons for the differences in vegetation productivity are indeed not only environmental factors, and the impact of human activities cannot be ignored. However, when we set up the transects, we have eliminated areas with obvious human activities through the field investigation in the early stage of the survey; the sites are mostly natural grasslands, and the productivity status reflected by them is mainly affected by factors such as climate and soil. After this revision, a description of "natural grassland" will also be added to the Method section (line: 156-157). 2) The response intensity is the slope of the linear regression. The greater the slope, the faster the rate of change of the region’s productivity with the environment, that is, the greater the response intensity. 3) We have modified [Fig 4] and completed the relevant conclusions and corrections of the discussion (line:258-260). 4) We have deleted the sentence to ensure the conciseness of the conclusion. 5) We have rewritten the title to summarize the selected information clearly and accurately (line: 274).

6. Discussion: generally, the discussion has poor relation with results obtained. The paper put environmental factors as fixed variable, while neglect their variations. Some conclusions lack of direct data results, while may be speculated (such as line 323).

Response：Thank you very much for your reminder. After this revision, we have added references to some of the arguments to improve the scientific rigor of the text. In addition, as a natural phenomenon (law), the change of environmental factors has been visualized in the results section, and its relationship with grassland productivity has also been demonstrated. Here we are more about discussing the possible reasons for this phenomenon and the law to appear in this way. We deleted some redundant text and rewritten subsections. And clarify the limitations of the comparative transect survey, that is, the potential method defects. However, we hope more to trigger discussion on ecological limiting factors at the regional scale, and in the regional integration analysis, we should pay attention to the problem that they may have different or even opposite response mechanisms.

7. Figure 1: there lack of statistical analysis for the three environmental factor limitations in columns. Figure 2: The ANPP increase exponentially with longitude on TP and LP? linearly on MP? correct or not? Figure 5: more information needed about the analysis?

Response：Thank you very much for your advice. [Figure 1]: We have completed the relevant significance analysis, based on Duncan's multiple calculations. [Figure 2]: To be precise, the curve fitting is drawn uniformly according to the polynomial (binomial); however, due to typesetting restrictions, the fitting equation is not indicated on the figure. At this time, the linear fitting of LP and TP reaches the maximum R^2; after this modification, MP will draw the exponential equation to have the maximum R^2. [Figure 5]: Using canonical correspondence analysis (CCA), taking the species biomass matrix of the site as the response variable and the environmental factor matrix as the explanatory variable, the interpretation (R^2) is obtained.

8. Table 1: the results in the table conflict with the hypothesis, or expectations. The data was significant with whom as marked with *?

Response：Thank the reviewers for their comments. The main master model in [Table 1] reflects the environmental regulation mechanism of vegetation productivity under the influence of major regional limiting factors. It uses backward filtering in stepwise regression, and considers the model with the smallest AIC value as the optimal model. The results are shown in Table 1; we believe that there is no conflict with the previous statement. Under the limitation of low temperature, the Tibetan Plateau has a relatively slow response rate to all environmental factors, and finally the master model of N element was screened out. Similar studies by others (line:356) have also corroborated this point, which will only remind us to pay more attention to the changes of N element in the Tibetan Plateau.  

To reviewer 2:

This study investigated the main regional limiting factors in the changes of grassland aboveground net primary productivity in Tibetan Plateau (TP), Mongolian Plateau (MP) and Loess Plateau (LP) by using a comparative transect survey dataset. This research topic is important as regional differences and regulation mechanism of vegetation productivity response to changing environmental is one of the core issues in macroecology research. To verify the main regional limiting factors, the authors used multiple analysis methods. However, analysis method usually has its own defects. For the analysis method of gradual regression, the significance depends not only on their intrinsic ecological implications but also on the variations of their selected samples. Why nutrient-N is the highest in TP, while the N relationship with TP productivity showed the best among the three regions. I think, at least partly, it is due to that the selected samples in TP have greater variations of N than those in the other two regions. Similarly, the variations of N in MP are much lower, thus it is difficult to develop a significant relationship between N and productivity. I suggest that the author fully acknowledged the limitations of this experimental design and clarified the potential methodological defects in the discussion. On the contrary, I like the results based on the canonical response analysis (CCA). When the data of the three regions are analyzed together, they found that the explanation degree of climate and soil nutrient factors reduced largely, indicating that the three plateaus may have different or even divergent productivity response mechanisms. I suggest that the author focus on this result for discussion. In addition, English writing could be improved. While the general writing was good, there were many Chinglish and long sentence that greatly affect the readability of this paper. For example, on Line 27, " MP is precipitation specific, LP is temperature and nutrient specific …"; Line 31" MP is mainly temperature and precipitation regulation, LP is temperature and nutrients, and TP is nutrients…". Lines 32-33, “The interpretation degree of environment to ANPP is 72.56% in the region, while only 27.18% after simple integration” Lines 341-345, “In the TP limited by low temperature, the response intensity…, which…,which…environmental factors in the TP.”

Based on the above reasons, I'd like to recommend a major revision before it can be accepted for publication.

Response：Thank you very much for the encouragement and comments of the reviewers. We fully agree with your opinion and made corresponding changes. In the Discussion, the deficiencies of the experimental method design in this article are explained (line:378-391). Pay attention to the different or even opposite productivity regulation mechanisms in different biota, perhaps the main limiting factor in the region can become a breakthrough point. In addition, we have revised the entire text one by one to avoid lengthy sentences.

Thanks again!

---

## [Editor Report · Decision Letter 1]

23 Sep 2020

Regional response of grassland productivity to changing environment conditions influenced by limiting factors

PONE-D-20-18058R1

Dear Dr. He,

We’re pleased to inform you that your manuscript has been judged scientifically suitable for publication and will be formally accepted for publication once it meets all outstanding technical requirements.

Kind regards,

Dafeng Hui, Ph.D.

Academic Editor

PLOS ONE

Additional Editor Comments (optional):

The authors have made efforts and addressed most of the reviewers' concerns.
---

## [Editor Report · Acceptance letter]

8 Oct 2020

PONE-D-20-18058R1 

Regional response of grassland productivity to changing environment conditions influenced by limiting factors 

Dear Dr. He:

I'm pleased to inform you that your manuscript has been deemed suitable for publication in PLOS ONE. Congratulations! Your manuscript is now with our production department. 

Kind regards, 

on behalf of

Dr. Dafeng Hui 

Academic Editor

PLOS ONE